# Population-based utility scores for HPV infection and cervical squamous cell carcinoma among Australian Indigenous women

**Xiangqun Ju**[1]*, **Karen Canfell**[2,3]**, Kirsten Howard**[3]**, Gail Garvey**[4]**, Joanne Hedges**[1]**, Megan Smith**[2,3]**, Lisa Jamieson**[1]

**1** Adelaide Dental School, Australian Research Centre for Population Oral Health, The University of Adelaide, Adelaide, Australia, **2** Cancer Council of NSW, Sydney, Australia, **3** School of Public Health, The University of Sydney, Sydney, Australia, **4** Menzies School of Health Research, Tiwi, Australia

* Xiangqun.ju@adelaide.edu.au

## Abstract

### Objective

Working in partnership with Indigenous communities in South Australia, we aimed to develop, pilot test and estimate utility scores for health states relating to cervical cancer screening, precancer, and invasive cervical cancer and precancer/cancer treatment among Indigenous women.

### Methods

Development and pilot testing of hypothetical cervical cancer health states, specifically through the lens of being an Indigenous Australian woman, was done with an Indigenous Reference Group in conjunction with five female Indigenous community members. Six health states were developed. These included: (1) Screened: cytology normal; (2) human papillomaviruses (HPV) positive with cytology normal; (3) low grade cytology (LSIL);(4) high grade cytology (HSIL); (5) early stage cervical cancer and; (6) later stage cervical cancer. Utility scores were calculated using a two-stage standard gamble approach among a large cohort of Indigenous Australian women taking part in a broader study involving oral HPV infection. The mean and standard deviation (SD) of the rank, percentage of respondents with a utility = 1 (perfect health) and utility score of each health state was summarised. Mean (SD) and medians and inter-quartile range (IQR) over 12 months and lifetime duration were calculated. Potential differences by age and residential location were assessed using the Wilcox Sum Rank test.

### Results

Data was obtained from 513 Indigenous women aged 19+ years. Mean utility scores were higher for the four non-cancer health states than for invasive cervical cancer states (p-values <0.05). Lower mean utility scores were observed for late stage cervical cancer, with

University of Adelaide Data Access (contact via Australian Research Centre for Population Oral Health: arcpoh@adelaide.edu.au) for researchers who meet the criteria for access to confidential data.

**Funding:** This study was funded by the Australia's National Health and Medical Research Council (APP1120215). The funder had no role in the design and conduct of the study; collection, management, analysis, and interpretation of the data; preparation, review, or approval of the manuscript; and decision to submit the manuscript for publication. LMJ had full access to all the data in the study and takes responsibility for the integrity of the data and the accuracy of the data analysis.

**Competing interests:** The authors confirm that they have no competing interests.

0.69 at 12 months and 0.70 for lifetime duration (Intra-class correlation coefficients = 0.425). Higher utility scores were observed for the four non-cancer health states among non-metropolitan participants (ranged from 0.93 to 0.98) compared with metropolitan participants (ranged from 0.86 to 0.93) (p-values<0.05).

## Conclusion

Among a large cohort of Indigenous Australian women, the reduction in quality of life (which utilities reflect) was perceived to be greater with increasing severity of cervical cancer health states. There were differences observed by geographic location, with positive cervical screening and precursor cancer-related quality of life being much higher among non-metropolitan-dwelling participants. These utility values, from one of the largest such studies ever performed in any population will be uniquely able to inform modelled evaluations of the benefits and costs of cervical cancer prevention interventions in Indigenous women.

## Introduction

Cervical cancer, the fourth most common cancer among women globally [1–3], is one of the major gynaecologic malignancies that threatens women's health and quality of life. The 5-year relative cervical cancer survival rate is generally higher in developed countries, including the United Kingdom (63.8%), Australia (67.1%), Denmark (69.5%) and Japan (71.4%) [4, 5] than in developing countries such as India (35%-60%), Chile (50.9%), South Africa (54.9%) and Brazil (61.1%) [5, 6]. In the vast majority of cases, cervical cancer, is caused by associated by human papillomavirus (HPV) infection [2, 7]. However, cervical cancer is a largely preventable disease through early and regular cervical screening (Pap Smears), HPV vaccination and timely treatment of precancerous lesions [2]. Due to lower levels of access to these proven interventions, in many high-income countries Indigenous women have a higher morbidity and mortality rate of cervical cancer than non-Indigenous women [8, 9].

Indigenous women in Australia are those identifying as Aboriginal and/or Torres Strait Islander. Indigenous women constitute approximately 3.2% of the total female Australian population according to the Australian Census 2016 [10]. While the incidence rate of cervical cancer decreased from 18 to 7 per 10,000 Australian women between 1990 and 2015, and Indigenous women had 2.5 times higher (22.2 per 10,000 women) the incidence rate and 3 times higher mortality rate than non-Indigenous women aged 20–69 years between 2011–2015 [11]. These differences are likely to be due, in part, lower access to cervical cancer screening and vaccination among Indigenous women [11–13].

Prevention of diseases is an effective way of reducing the burden of cervical cancer over the long term. Australia's National Cervical Screening Program was established in 1991 and renewed in 2017 (including renew cervical screening from 2 to 5 years, age-range from 25 to 74 years old, and self-collection of a vaginal sample based on primary test for oncogenic type of HPV [14, 15], with the HPV vaccination via school-based immunisation program introduced for girls in 2007 [16]. In general terms, evaluation of the benefits, harms and cost-effectiveness of strategies for primary and secondary cancer prevention require an understanding of health state utilities. Health state utilities capture a given population's preferences for a range of health states. Although cervical cancer health state utilities have been assessed in several populations [17], the development of appropriate descriptions of the associated health

states and their assessment in Indigenous women, in Australia or any other country, has not been described. The aims of this study were to thus, in partnership with Indigenous communities in South Australia, develop, pilot test and estimate utility scores for health states relating to cervical cancer screening, precursor cervical cancer, established cervical cancer and its treatment among Indigenous women aged 18 years and over.

## Methods

This study developed six hypothetical health states from ten cervical cancer health states, then conducted pilot testing. A two-stage standard gamble (SG) approach was used to assess the health states. SG is grounded in expected utility theory, and often viewed as a gold standard to measure health utility [18].

### Health state development

In partnership with the study's Indigenous Reference Group (IRG), which comprised several respected Indigenous adults with diverse backgrounds from across South Australia, ten hypothetical cervical cancer prevention and treatment health state scenarios were developed. The scenario descriptions were informed by relevant Australian cervical cancer screening and treatment guidelines, including the psychosocial literature in relation to Indigenous health, and incorporated the feedback from the IRG [17, 19, 20]. Domains that the IRG identified as being of fundamental importance in the health state preferences included: 1) racism/distrust/confusion of health sector (with anticipation of racism being very strong); 2) connection/responsibilities to family; 3) social determinants of health uniquely over represented in many Indigenous families (death, incarceration, child removal from family by state, poverty, domestic violence, addictions, food insecurity, loss and grief, hum-bugging (concept of 'what's yours is mine'); 4) connections with country (especially for remote-dwelling participants) and; 5) spiritual thought processes; acceptance that sickness is their lot, accepting cancer is being 'sung to death'; going to the ancestors. The ten hypothetical health states included: 1) pap smear cytology normal; 2) HPV vaccination; 3) low-grade cytology; 4) low-grade cytology with colposcopy normal; 5) HPV positive with cytology normal; 6) HPV positive with colposcopy normal; 7) treated genital warts; 8) high grade cytology with histologically-confirmed Grade I cervical intraepithelial neoplasia; 9) high grade cytology with histologically-confirmed Grade II/III cervical intraepithelial neoplasia and; 10) early stage cervical cancer. After pilot testing, it became apparent that the burden of valuing 10 health states in terms of the time taken to complete with a two-stage standard gamble was significant. The number of health states was thus reduced to six (S1 File): 1) Screened: cytology normal; 2) HPV-positive with cytology normal; 3) low grade cytology; 4) high grade cytology; 5) early grade cervical cancer and; 6) late stage cervical cancer.

### Data collection

The IRG considered it imperative to hire and train research officers who were able to respectfully engage in, and be responsive to, the cultural values of participants, so that participants felt comfortable during the interview and to ensure interviewer consistency. This was achieved by the IRG being actively involved in the recruitment and training of research officers. Valuations of the six health states were collected from 513 female Indigenous Australians aged 19+ (ranging from 19 to 78) years, residing in South Australia and taking part in a wider study examining oral HPV infection and oropharyngeal cancer in Australia [21]. Participants had been initially recruited through Aboriginal Community Controlled Health Organisations (ACCHOs), who were key stakeholders in the study between Feb 2018 and January 2019 [21,

22]. Data for the cervical cancer health state utilities was collected during the parent study's 12-month follow-up (suspended early because of covid-19 restrictions) from February 2019 to March 2020. Participants not enrolled during the original recruitment period will not be eligible to participate in the follow-up phases.

**Ethics approval and consent to participate.**   Ethical approval was obtained from the University of Adelaide Human Research Ethics Committee (H-2016-246) and the Aboriginal Health Council of South Australia (04-17-729). All participants provided signed informed consent.

**Health state collection.**   Data from the six hypothetical HPV-related cervical cancer health states were collected via face-to-face interviews by trained research officers. Each scenario was described in a narrative format with the use of visual prompts and aids. Participants were invited to ask as many questions as they liked for clarification purposes. Participants were then asked to rank the description of each health state relative to the others, from one to six (one being most desirable, six being least desirable; equal ranking was accepted). At no stage were participants asked if they, or anyone they knew, had experienced any of the health states. The participant interviews averaged one hour in duration (ranging from 45 minutes to one hour 20 minutes). This was critical to ensure participants both understood the scenarios and were able to provide a meaningful and accurate portrayal of how participants framed the health states.

**Health state assessment.**   A two-stage standard gamble approach (Fig 1) was utilised [17, 23]. This was to enable a 'disutility' of a health state to be calculated by observing an individual's willingness to accept a certain risk of death in order to avoid the state. Participants were asked to imagine the five-temporary health states (from non-cancer to early cervical cancer) returning to full health after 12 months (Stage 1). For the late stage cervical cancer health state, where the probability of indifference between living with late stage cervical cancer is measured relative to the risky prospects associated with either perfect health or immediate death (Stage 2), two 'time in state' durations were used. The first was for 12 months followed by sudden and painless death, the second was from the present until age 85 years, followed by sudden and painless death. Participants therefore provided seven health state preference scores; five for each temporary health state and two for late stage cervical cancer.

Utility scores were determined for the temporary health states on a 0–1 cardinal interval scale, and mathematically transformed using the following function:

$$h_i = p_i + (1 - p_i) h_k$$

where $h_i$ is the utility of the temporary health state, $P_i$ is the probability of indifference observed between the certain outcome of experiencing the temporary health state and the risky prospect of either living with late stage cervical cancer or living with perfect health. $h_k$ is the utility of late stage cervical cancer (worst health outcome) evaluated on the death to perfect health scale (17). For late stage cervical cancer there are thus two $h_k$; one valued on the 12-month time scale, the other valued on the lifetime scale. For each individual utility score of a temporary health state we applied two separate 'time in state' values representing the anchor state, using the mathematical function above. Higher utility scores indicate the more preferred health states [24].

**Demographic data.**   In addition to the standard gamble exercise, data on participants' demographic characteristics were collected. Age was dichotomised into '18–40 years' and '>40 years' (median age was 40.0 years old), while residential location was defined as 'Metropolitan' and 'Non-metropolitan'.

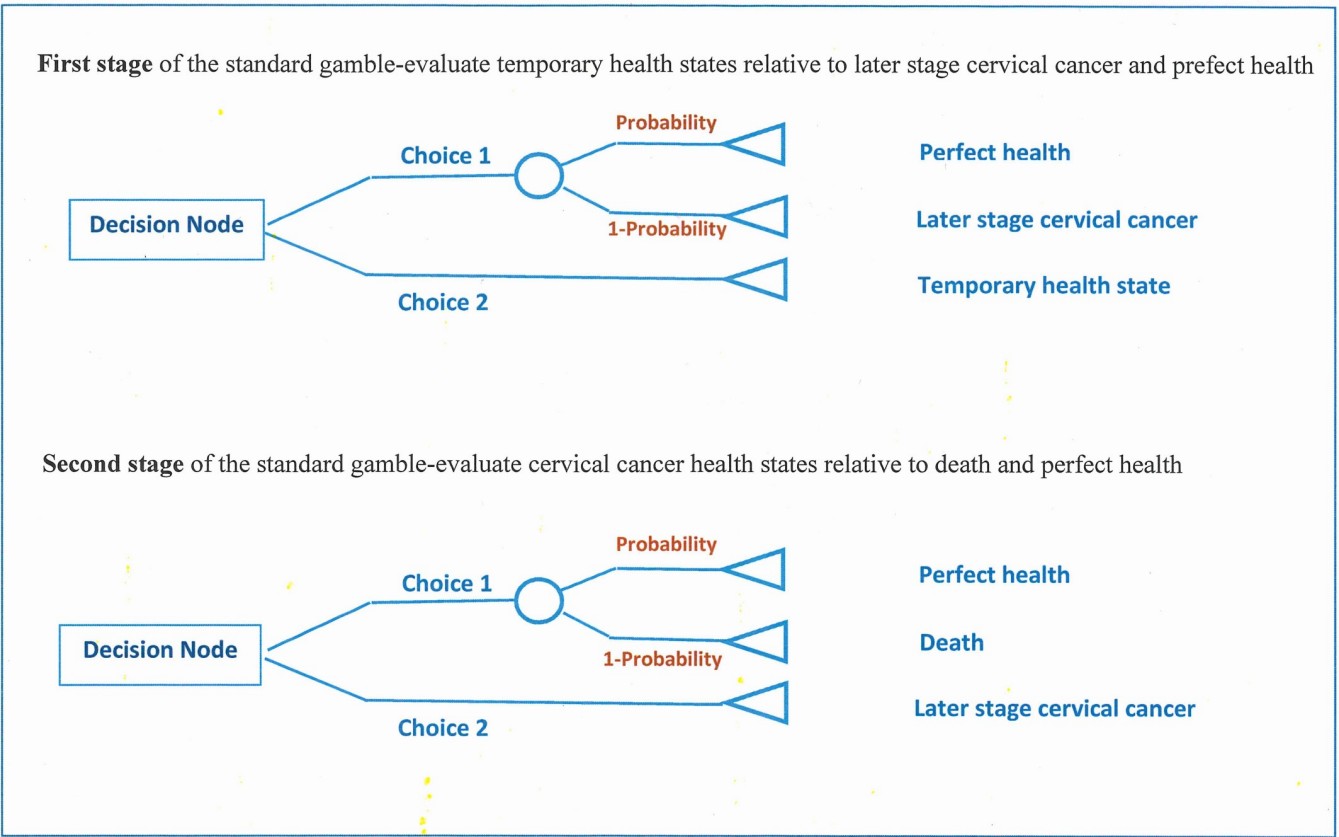

**Fig 1. Decision tree of two-stage standard gamble utilities.**

### Statistical analysis

Demographic characteristics were described by number and percentage. Each health state was ranked, with the percentage of perfect health at 12 months estimated. The utility score of each health state was summarised using means and standard deviation (SD), as well as medians and inter-quartile range (IQR). Intra-class correlation coefficients (ICC) were used to assess the level of agreement between the pair of 'late stage cervical cancer' scores valued using '12 months' and 'lifetime' durations ('time in state' values) [17].

ICC was evaluated with the following formula:

$$ICC = \frac{MS_r - MS_e}{MS_r + (k-1)MS_e + \frac{k}{n}(MS_c + MS_e)}$$

Where **MS** is mean square, **MS_r** is the difference between the grand mean for a health state (combining both sets of utility scores calculated using different 'time in state' values) and the group means for a health state calculated according to a specific 'time in state' value. **MS_c** is characterised as the difference between the specific 'time in state' individual utility scores and the mean of these scores. **MS_e** is mean square for error, **n** is the number of participants and **k** is measurement value. To enable calculation of ICCs, two-way mixed effects, absolute agreement and single measurement were carried out for each health state, such that mean square values ('between groups' and 'within groups') were determined for utility scores based on each 'time in state' anchor point [17, 25]. ICC values above 0.90 were considered 'excellent', between

0.75–0.90 and 0.50–0.75 were 'good' and 'moderate' agreement respectively, while those less than 0.50 were considered 'poor' [25].

The Wilcoxon Sum Rank test was used to assess differences in mean utility scores for each temporary health state according to age and residential location [26]. Utility scores transformed with '12 month' and 'lifetime' durations were analysed separately for each demographic outcome. Given that six distinct health states were being tested, with two sets of 'time in state' anchor states for the mathematical transformation of temporary health states, a total of 12 tests for statistical significance were made.

SAS statistical software (SAS 9.4, SAS Institute Inc., Cary, NC, USA) was used to analyse the data.

## Results

A total of 513 Indigenous Australian women residing in South Australia, aged 19 to 78 years, completed the cervical cancer health state utilities questionnaire. The average age was 41.8 years, with more than 50 percent aged 40 years or older. Nearly two-thirds resided in non-metropolitan locations (see Table 1).

The ordinal rank of each health state, and percentage of perfect health on each health state lasting 12 months, is presented in Table 2. The highest ranked was 'Cervical screened, cytology normal' (median rank: 1; IQR: 1–2), with an average of 85 percent of perfect health lasting 12 months. The lowest ranked was 'late stage cervical cancer'.

Table 2 presents the mean and SD, and median and IQR; distributions are presented in Figs 2 and 3 for utility scores for each health state at '12 months' and 'lifetime' duration, respectively. Mean utility scores were higher for the four non-cancer health states than for 'early and late stage cervical cancer' ('0.89–0.98' vs '0.69–0.84'). Lower mean utility scores were observed for 'late stage cervical cancer' with 0.69 at 12 months and 0.70 for lifetime duration. Meanwhile, the interquartile range for the four non-cancer and early stage cervical cancer health states was '1.00 to 1.00' when anchored to both 12-month and lifetime duration, compared with, for 'late stage cervical cancer', 0.00–1.00 for both 12-month and lifetime duration. This indicated heterogeneity in the valuation of late stage cervical cancer compared to other health states.

The intraclass correlation coefficient between results which anchored to '12-months' versus 'lifetime duration' for 'late stage cervical cancer' was 0.76 to 0.79 for most health states, except the best health state: 'cytology normal (ICC:0.48)' and the worst health state: 'late stage cervical cancer (ICC: 0.43)' (Table 2). This indicates good agreement for most health state utility scores assessed using a '12-month' and 'lifetime' duration.

The utility scores based on anchoring to the '12-month' and 'lifetime' duration late stage cervical cancer stratified by demographic characteristics are shown in Table 3. There were no

**Table 1. Sample characteristics among Indigenous women aged 18+ years.**

|  | Number | Percentage (95% CI) |
|---|---|---|
| **Total** | 513 | 100.0 |
| **Age groups (years)** |  |  |
| < 40 | 249 | 48.5 (44.2–52.9) |
| ≥ 40 | 264 | 51.5 (47.1–55.8) |
| **Location** |  |  |
| Metro | 195 | 38.0 (33.8–42.2) |
| regional | 318 | 62.0 (57.8–66.2) |

**Table 2. Hypothetical health state scenarios relating to genital HPV infection and cervical cancer, ranks and percentages of perfect health by six health states lasting 12 months, standard gamble utility scores and intra-class correlation coefficient for cervical cancer duration among Indigenous women.**

| Codes | Health states | Description | Rank | Perfect health lasting 12 months (%) | | Cervical cancer duration | | | | |
| | | | | | | 12 months | | Lifetime | | |
| | | | Median (IQR) | Mean (SD) | Median (IQR) | Mean (SD) | Median (IQR) | Mean (SD) | Median (IQR) | ICC |
|---|---|---|---|---|---|---|---|---|---|---|
| S1 | Screened; cytology normal | Cervical screening test cytology negative (Moira) | 1 (1–1) | 92.1(13.8) | 100.0 (90.0–100.0) | 0.95 (0.23) | 1.00 (1.00–1.00) | 0.95 (0.21) | 1.00 (1.00–1.00) | 0.47610 |
| S2 | HPV positive with cytology normal | HPV positive and cytology negative; follow-up cervical screening in 12 months (Agnes) | 2 (2–3) | 78.6(18.0) | 80.0 (70.0–90.0) | 0.92 (0.26) | 1.00 (1.00–1.00) | 0.93 (0.26) | 1.00 (1.00–1.00) | 0.76353 |
| S3 | Low grade cytology | Cytology screening with a low-grade abnormality, follow-up cervical screening in 12 months (Vera) | 3 (3–4) | 72.9 (17.4) | 76.0 (60.0–85.0) | 0.90 (0.29) | 1.00 (1.00–1.00) | 0.91 (0.29) | 1.00 (0.99–1.00) | 0.75126 |
| S4 | High grade cytology | Cytology screening with a high-grade abnormality requiring treatment to remove abnormal cells (Gina) | 4 (3–4) | 65.3 (19.0) | 70.0 (50.0–80.0) | 0.90 (0.30) | 1.00 (1.00–1.00) | 0.89 (0.31) | 1.00 (0.99–1.00) | 0.77777 |
| S5 | Early stage cervical cancer | Early stage cervical cancer requiring a hysterectomy (Allison) | 5 (5–5) | 49.2 (26.5) | 50.0 (30.0–70.0) | 0.83 (0.37) | 1.00 (1.00–1.00) | 0.84 (0.37) | 1.00 (0.99–1.00) | 0.79778 |
| S6 | Late stage cervical cancer | Late stage cervical cancer requiring chemotherapy and radiation therapy, and ongoing monitoring visit (Celeste) | 6 (6–6) | 33.5 (23.8) | 30.0 (10.0–50.0) | 0.69 (0.46) | 1.00 (0.00–1.00) | 0.70 (0.45) | 1.00 (0.00–1.00) | 0.42555 |

Notes: HPV: human papillomavirus; Interquartile range (IQR) being equal to the difference between 75th and 25th percentiles.

significant differences in mean utility scores between age groups for each health state, either when anchored to 12-months or lifetime duration.

Participants residing in non-metropolitan locations had significantly higher utility scores for 'Screened: cytology normal' (0.97 vs 0.93) and 'HPV positive with cytology normal' (0.95 vs

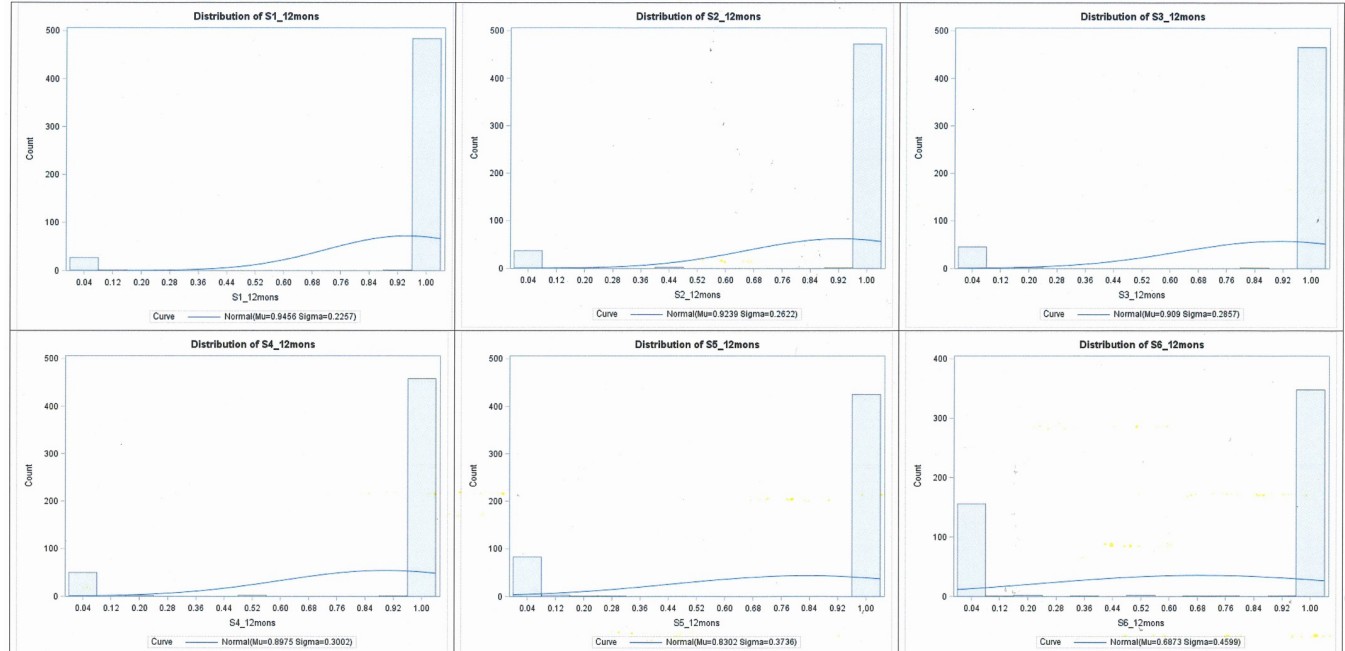

**Fig 2. Standard gamble utility score distributions at the 12 months late stage cervical cancer.** Legend: S1: Screened; cytology normal; S2: HPV positive with cytology normal; S3: Low grade cytology; S4: High grade cytology; S5: Early stage cervical cancer and S6: Late stage cervical cancer.

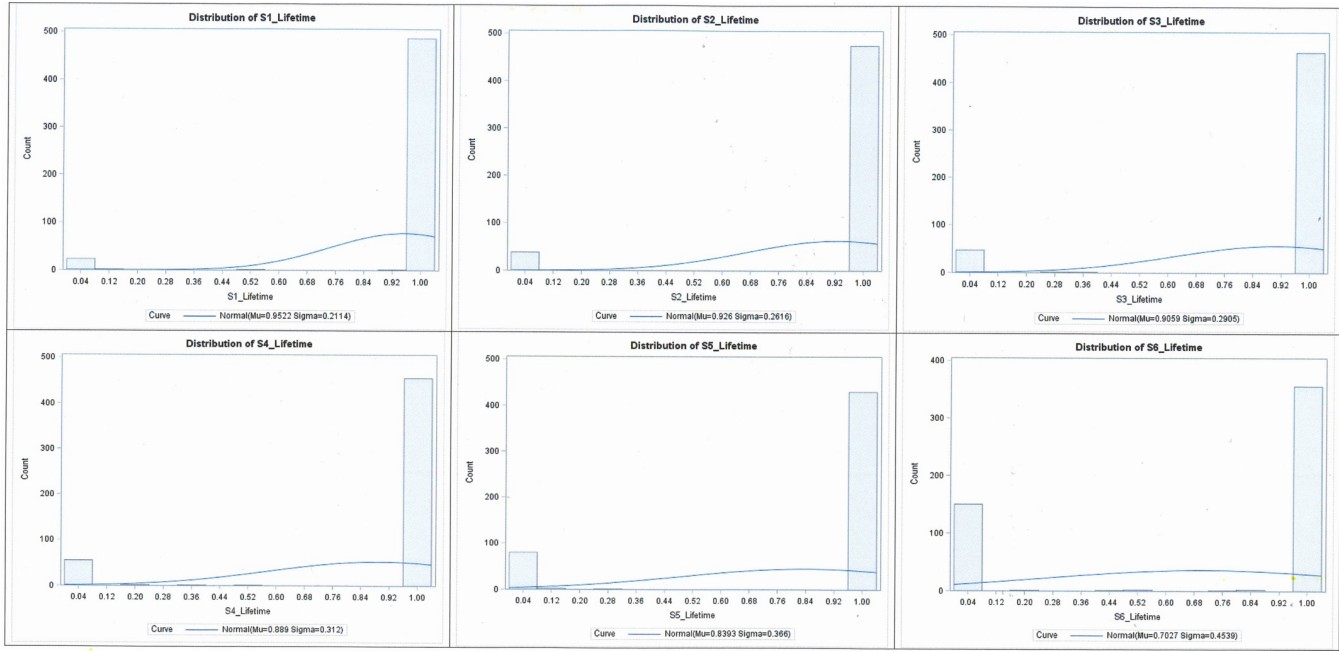

**Fig 3. Standard gamble utility score distributions at the lifetime duration late stage cervical cancer.** Legend: S1: Screened; cytology normal; S2: HPV positive with cytology normal; S3: Low grade cytology; S4: High grade cytology; S5: Early stage cervical cancer and S6: Late stage cervical cancer.

0.91) than those residing in metropolitan locations at 12 months. At lifetime duration, participants residing in non-metropolitan locations had significantly higher utility scores for four non-cancer health states than their counterparts, which were '0.98 vs 0.93', '0.95 vs 0.91', 0.94 vs 0.88' and '0.93 vs 0.85', respectively. There were no significant differences in mean utility scores between the two location groups for both early and late stage cervical cancer health states, either when anchored to 12-months or lifetime duration.

## Discussion

Working in partnership with Indigenous communities in South Australia, this is the first study to develop, pilot test and estimate utility values for hypothetical cervical cancer health states from the perspectives of Indigenous Australian women. These utility values, from one of the largest such studies ever performed in any population will be uniquely able to inform modelled evaluations of the benefits and costs of cervical cancer prevention interventions in Indigenous women. Our findings demonstrated that both early and late cervical cancer health states were associated with lower utility scores, irrespective of the length of duration of the anchor state, compared with the four other non-cancer health states. The positive impact of hypothetical cervical screening and precursor cervical cancer on quality of life were much greater for Indigenous women living in non-metropolitan areas compared to those in metropolitan locations. Our findings indicate broadly lower utility scores for cervical cancer screening, abnormalities and cervical cancer health states among Indigenous Australian women compared with general population estimates of non-Indigenous Australian women [17].

Cervical cancer screening is an important and cost-effective strategy of reducing the prevalence and severity of cervical cancer. Comparing with a similar predicate study [17] with 10 cervical cancer health status among all Australian population, our findings suggest that Indigenous women may be less likely to participate in screening (Utility score: 0.95) than non-

**Table 3. Age and residential location comparisons of standard gamble utility scores using the '12 month' and 'lifetime' duration later stage cervical cancer.**

| | | Age groups (years) | | | | | Location | | | | |
| | | 18–40 | | >40 | | | Metropolitan | | Non-metropolitan | | |
| | Health states (codes) | Mean (SD) | Median (IQR) | Mean (SD) | Median (IQR) | P-value | Mean (SD) | Median (IQR) | Mean (SD) | Median (IQR) | P-value |
|---|---|---|---|---|---|---|---|---|---|---|---|
| **12 month–late stage cervical cancer** | S1 | 0.94 (0.24) | 1.00 (1.00–1.00) | 0.95 (0.22) | 1.00 (1.00–1.00) | 0.3847 | 0.93 (0.28) | 1.00 (1.00–1.00) | 0.97 (0.16) | 1.00 (1.00–1.00) | 0.0353 |
| | S2 | 0.92 (0.26) | 1.00 (1.00–1.00) | 0.92 (0.26) | 1.00 (1.00–1.00) | 0.7333 | 0.91 (0.28) | 1.00 (1.00–1.00) | 0.95 (0.22) | 1.00 (1.00–1.00) | 0.0347 |
| | S3 | 0.90 (0.29) | 1.00 (1.00–1.00) | 0.91 (0.28) | 1.00 (1.00–1.00) | 0.5505 | 0.90 (0.29) | 1.00 (0.99–1.00) | 0.92 (0.27) | 1.00 (1.00–1.00) | 0.4245 |
| | S4 | 0.90 (0.30) | 1.00 (1.00–1.00) | 0.90 (0.30) | 1.00 (1.00–1.00) | 0.9826 | 0.88 (0.32) | 1.00 (1.00–1.00) | 0.92 (0.26) | 1.00 (1.00–1.00) | 0.1286 |
| | S5 | 0.83 (0.38) | 1.00 (1.00–1.00) | 0.84 (0.37) | 1.00 (1.00–1.00) | 0.7894 | 0.84 (0.36) | 1.00 (1.00–1.00) | 0.80 (0.40) | 1.00 (1.00–1.00) | 0.2479 |
| | S6 | 0.68 (0.46) | 1.00 (0.00–1.00) | 0.70 (0.46) | 1.00 (0.00–1.00) | 0.4472 | 0.70 (0.46) | 1.00 (0.00–1.00) | 0.66 (0.47) | 1.00 (0.00–1.00) | 0.4698 |
| **Lifetime duration–late stage cervical cancer** | S1 | 0.94 (0.23) | 1.00 (1.00–1.00) | 0.94 (0.23) | 1.00 (1.00–1.00) | 0.2624 | 0.93 (0.25) | 1.00 (1.00–1.00) | 0.98 (0.12) | 1.00 (1.00–1.00) | 0.0174 |
| | S2 | 0.93 (0.26) | 1.00 (1.00–1.00) | 0.93 (0.26) | 1.00 (1.00–1.00) | 0.9169 | 0.91 (0.29) | 1.00 (1.00–1.00) | 0.95 (0.21) | 1.00 (1.00–1.00) | 0.0296 |
| | S3 | 0.91 (0.28) | 1.00 (0.99–1.00) | 0.90 (0.30) | 1.00 (1.00–1.00) | 0.6113 | 0.88 (0.32) | 1.00 (1.00–1.00) | 0.94 (0.24) | 1.00 (1.00–1.00) | 0.0194 |
| | S4 | 0.89 (0.31) | 1.00 (1.00–1.00) | 0.89 (0.33) | 1.00 (1.00–1.00) | 0.4944 | 0.86 (0.31) | 1.00 (1.00–1.00) | 0.93 (0.25) | 1.00 (1.00–1.00) | 0.0148 |
| | S5 | 0.85 (0.35) | 1.00 (1.00–1.00) | 0.83 (0.38) | 1.00 (1.00–1.00) | 0.3836 | 0.85 (0.35) | 1.00 (1.00–1.00) | 0.82 (0.39) | 1.00 (1.00–1.00) | 0.5466 |
| | S6 | 0.70 (0.46) | 1.00 (0.00–1.00) | 0.71 (0.45) | 1.00 (0.00–1.00) | 0.7834 | 0.70 (0.45) | 1.00 (1.00–1.00) | 0.70 (0.45) | 1.00 (0.00–1.00) | 0.9106 |

Notes: Interquartile range (IQR) being equal to the difference between 75th and 25th percentiles. S1: Screened; cytology normal; S2: HPV positive with cytology normal; S3: Low grade cytology; S4: High grade cytology; S5: Early stage cervical cancer and S6: Late stage cervical cancer.

Indigenous women (Utility score: 0.99) in Australia. The trend of cervical cancer screening in our study is consistent with other practical studies in Indigenous populations at a global level, such as the First Nations people in Canada [27], the Indigenous Peruvian women in Peru [28] and American Indian women in the United States [29].

Negative attitudes about cancer treatment lead to decreased survival of cervical cancer [30]. Our findings indicated that Indigenous women had lower utility scores for early- (0.83–0.84), and late-stage cervical cancer (0.69–0.70) than their non-Indigenous counterparts (0.97–0.99) [17] at both 12-months and lifetime duration. The findings are similar to a study involving American Indian women, who had a lower cancer survival rate than white women due to lower level of basic cancer screening knowledge and more negative attitudes about cancer treatment [29].

It is interesting that higher scores were observed for the four non-cancer health states among non-metropolitan Indigenous women in our study (utility scores ranged from 0.93 to 0.98), compared with metropolitan participants (utility scores ranged from 0.86 to 0.93). The findings are certainly contrary to other cancer estimates among Indigenous populations in Australia, where higher burden of cancer and lower cervical screening are evidenced among those living in regional or remote locations compared to those living in urban areas [12, 31]. However, a current study [32] has shown that the proportion of women cervical cancer screening was higher for the Indigenous Primary Health Care (PHC) centres in the very remote

areas than for centres in other areas. Our findings may associate with the utility of specific models that PHC centres in very remote areas have implemented in order to improve quality of life.

The strengths of this study include: 1) the extensive development, pilot testing and refinement of the health states conducted through an Indigenous Reference Group. This is considered essential in any stage of developing tools for use with and by Indigenous Australians; 2) the health states were then tested in a large sample of Indigenous South Australian women. The study had extremely good Indigenous community buy-in, with some participants going to considerable length to contact the research team to enquire what they needed to do to be involved; 3) a two-stage standard gamble approach was used to yield helpful data for cost effective and health economic analysis. Limitations include participants being based in South Australia only, meaning the findings may not be generalisable to the many other culturally and linguistically diverse Indigenous groups elsewhere in Australia. The findings, for the same reason, may also not be generalisable to other Indigenous groups in the world.

## Conclusion

We used culturally respectful processes to develop six cervical cancer health states representing cervical screening, HPV infection, screen-detected abnormalities, and early and late cervical cancer from the perspective of Indigenous Australian women. The reduction in quality of life was perceived to be greater with increasing severity of health states. There were differences observed by geographic location, with positive cervical screening and precursor cancer-related quality of life being much higher among non-metropolitan participants. Our findings are an important contribution to cost-utility and disease prevention strategies that seek to inform policies around reducing HPV infection and cervical cancer among all Australian women. The information could be used to directly calculate quality-adjusted life years and to, in turn, be translated into health policy regarding Indigenous patient journeys with primary and secondary prevention for HPV-related cervical cancer.

## Supporting information

**S1 File. Health state vignettes.**
(PDF)

## Acknowledgments

This study was governed by an Indigenous Reference Group, who oversaw the orchestration, delivery and feedback of the study findings as it relates to the health and well-being of Indigenous Australians. We sincerely acknowledge and appreciate all that this Reference Group did. We also thank and acknowledge all study participants, and the staff who collected data.

## Author Contributions

**Conceptualization:** Karen Canfell, Lisa Jamieson.

**Data curation:** Xiangqun Ju, Gail Garvey, Joanne Hedges, Lisa Jamieson.

**Formal analysis:** Xiangqun Ju.

**Funding acquisition:** Lisa Jamieson.

**Investigation:** Karen Canfell, Gail Garvey, Joanne Hedges, Megan Smith, Lisa Jamieson.

**Methodology:** Xiangqun Ju, Karen Canfell, Kirsten Howard, Megan Smith, Lisa Jamieson.

**Project administration:** Karen Canfell, Joanne Hedges, Lisa Jamieson.

**Software:** Xiangqun Ju.

**Writing – original draft:** Xiangqun Ju.

**Writing – review & editing:** Xiangqun Ju, Karen Canfell, Kirsten Howard, Gail Garvey, Joanne Hedges, Megan Smith, Lisa Jamieson.

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
