## [Decision Letter · Decision Letter 0]

18 Jan 2021

PONE-D-20-26869

Population-based utility scores for HPV infection and cervical squamous cell carcinoma among Australian Indigenous women

PLOS ONE

Dear Dr. Ju,

Thank you for submitting your manuscript to PLOS ONE. After careful consideration, we feel that it has merit but does not fully meet PLOS ONE’s publication criteria as it currently stands. Therefore, we invite you to submit a revised version of the manuscript that addresses the points raised during the review process.

We look forward to receiving your revised manuscript.

Kind regards,

John S Lambert

Academic Editor

PLOS ONE

Journal Requirements:

2. Data availability. Please note that PLOS journals require authors to make all data underlying the findings described in their manuscript fully available without restriction, with rare exception. Unfortunately, your statement that "No- some restrictions will apply", is not in accordance with PLOS data availability policy. PLOS requires that a “minimal data set” is shared, defined as the data set used to reach the conclusions drawn in the manuscript with related metadata and methods, and any additional data required to replicate the reported study findings in their entirety. Authors do not need to submit their entire data set if only a portion of the data were used in the reported study. Also, authors do not need to submit the raw data collected during an investigation if the standard in the field is to share data that have been processed. Please submit the following data: The values behind the means, standard deviations and other measures reported; The values used to build graphs; The points extracted from images for analysis.” http://journals.plos.org/plosone/s/data-availability#loc-faqs-for-data-policy. If you are unable to share the data, this may result in manuscript rejection.

4. We note that your paper includes detailed descriptions of individual patients/participants. As per the PLOS ONE policy (http://journals.plos.org/plosone/s/submission-guidelines#loc-human-subjects-research) on papers that include identifying, or potentially identifying, information, the individual(s) or parent(s)/guardian(s) must be informed of the terms of the PLOS open-access (CC-BY) license and provide specific permission for publication of these details under the terms of this license. Please download the Consent Form for Publication in a PLOS Journal (http://journals.plos.org/plosone/s/file?id=8ce6/plos-consent-form-english.pdf). The signed consent form should not be submitted with the manuscript, but should be securely filed in the individual's case notes. Please amend the methods section and ethics statement of the manuscript to explicitly state that the patient/participant has provided consent for publication: “The individual in this manuscript has given written informed consent (as outlined in PLOS consent form) to publish these case details”.

Reviewers' comments:

Reviewer's Responses to Questions

**Comments to the Author**

1. Is the manuscript technically sound, and do the data support the conclusions?

Reviewer #1: Yes

2. Has the statistical analysis been performed appropriately and rigorously? 

Reviewer #1: Yes

3. Have the authors made all data underlying the findings in their manuscript fully available?

Reviewer #1: Yes

4. Is the manuscript presented in an intelligible fashion and written in standard English?

Reviewer #1: Yes

5. Review Comments to the Author

Reviewer #1: - Abstract: in results part should be added the statistical tests results in addition to description presentation.

- Methods: It should be explained why among the health utility measurement instruments, the single measurement instrument is choosen and why Standard Gamble is preferred in this study instead of other instruments.

- Methods: What are the inclusion and exclusion citeria of the study participants?

6. PLOS authors have the option to publish the peer review history of their article (what does this mean?). If published, this will include your full peer review and any attached files.

Reviewer #1: No

---

## [Author Response · Author response to Decision Letter 0]

13 Feb 2021

Reviewer #1: 

1. Abstract: in results part should be added the statistical tests results in addition to description presentation.

Thank you. We added the statistical tests results, such as p-values and Intra-class correlation coefficients, in the Results section (line 53-58, page 4).

2. Methods: It should be explained why among the health utility measurement instruments; the single measurement instrument is chosen and why Standard Gamble is preferred in this study instead of other instruments.

We added a sentence to explain why the ‘Standard Gamble’ as a single measurement instrument was chosen:

1) ‘Standard Gamble is grounded in expected utility theory, and often viewed as a gold standard to measure health utility.’ (line 113-115, page 6). 

3. Methods: What are the inclusion and exclusion criteria of the study participants?

We added sentences to introduce the inclusion and exclusion criteria of the study participants:

1) ‘Valuations of the six health states were collected from 513 female Indigenous Australians aged 19+ (ranging from 19 to 78) years, residing in South Australia and taking part in a wider study examining oral HPV infection and oropharyngeal cancer in Australia.’ (line 147-150, page 8).

2) ‘Participants not enrolled during the original recruitment period will not be eligible to participate in the follow-up phases.’ (line 155-156, page 8).

---

## [Decision Letter · Decision Letter 1]

15 Jun 2021

PONE-D-20-26869R1

Population-based utility scores for HPV infection and cervical squamous cell carcinoma among Australian Indigenous women

PLOS ONE

Dear Dr. Ju,

Thank you for submitting your manuscript to PLOS ONE. After careful consideration, we feel that it has merit but does not fully meet PLOS ONE’s publication criteria as it currently stands. Therefore, we invite you to submit a revised version of the manuscript that addresses the points raised during the review process.

Please revise the manuscript to address all the reviewer's comments in a point-by-point response in order to ensure it is meeting the journal's publication criteria. Please note that the revised manuscript will need to undergo further review, we thus cannot at this point anticipate the outcome of the evaluation process.

We look forward to receiving your revised manuscript.

Kind regards,

Miquel Vall-llosera Camps

Senior Editor

PLOS ONE

Journal Requirements:

Reviewers' comments:

Reviewer's Responses to Questions

**Comments to the Author**

1. If the authors have adequately addressed your comments raised in a previous round of review and you feel that this manuscript is now acceptable for publication, you may indicate that here to bypass the “Comments to the Author” section, enter your conflict of interest statement in the “Confidential to Editor” section, and submit your "Accept" recommendation.

Reviewer #2: All comments have been addressed

Reviewer #3: (No Response)

Reviewer #4: (No Response)

2. Is the manuscript technically sound, and do the data support the conclusions?

Reviewer #2: Yes

Reviewer #3: Yes

Reviewer #4: Yes

3. Has the statistical analysis been performed appropriately and rigorously? 

Reviewer #2: Yes

Reviewer #3: I Don't Know

Reviewer #4: Yes

4. Have the authors made all data underlying the findings in their manuscript fully available?

Reviewer #2: (No Response)

Reviewer #3: Yes

Reviewer #4: No

5. Is the manuscript presented in an intelligible fashion and written in standard English?

Reviewer #2: Yes

Reviewer #3: Yes

Reviewer #4: Yes

6. Review Comments to the Author

Reviewer #2: (No Response)

Reviewer #3: I do not have the expertise to judge the specific tools used for this study however I note that the study has followed proper process for engaging with Indigenous populations, which speaks to its success overall. I felt it read clearly and logically and only have minor comments/corrections.

Comment:

Short title: Needs to include the word 'Indigenous' as that is a major focus

Minor corrections:

Line 105:take out one ‘in’ before Indigenous

Line 112: should the word ‘status’ actually be ‘states’ (appears twice in this line)

Line 113: change conduced to ‘conducted’

Line 198: separates should be ‘separate’ (singular)

Line 282: “each health state” not states

Line 328: add word to after similar- “similar to a study”

Line 360: Indigenous Australian women (no ‘s’ necessary on Australian)

Reviewer #4: The authors have addressed a significant area of research and this article will contribute significantly towards the existing literature for cervical cancer prevention program evaluation with regards to health services uptake among indigenous communities in Australia. Yet there are some points that need to be considered:

1) Minor confusion exists in the abstract (Methods) line 40-41. Difference is not obvious between health states 2 and 3. Whereas it has been clearly indicated between S2 (HPV positive with cytology normal) and S3 (Low grade cytology) in the results section.

2) The introduction section effectively gives the background of the research problem but careful proofreading would be beneficial for language errors. Similarly the discussion section has discussed the results in comparison to previous studies, yet some language errors are present.

3) In the data collection section, the study referenced for description of sample size (19) does not explain the participants. The references need to be rearranged correctly.

4) Statistical analytical techniques used are adequate, tables have been formed precisely and the main findings reported appropriately. However with my limited expertise in the field of Health Economics, I am unable to advise much.

7. PLOS authors have the option to publish the peer review history of their article (what does this mean?). If published, this will include your full peer review and any attached files.

Reviewer #2: No

Reviewer #3: No

Reviewer #4: No

---

## [Author Response · Author response to Decision Letter 1]

17 Jun 2021

Response to reviewers

Review Comments to the Author

Reviewer #2: All comments have been addressed

Thanks.

Reviewer #3: 

1. I do not have the expertise to judge the specific tools used for this study however I note that the study has followed proper process for engaging with Indigenous populations, which speaks to its success overall. 

We appreciate this comment, thank you.

2. I felt it read clearly and logically and only have minor comments/corrections:

1). Short title: Needs to include the word 'Indigenous' as that is a major focus

We have changed short title to: ‘Utility scores for HPV infection and cervical squamous cell carcinoma among Australian Indigenous women’

2) corrections:

(1). Line 105: take out one ‘in’ before Indigenous

We have deleted ‘in’ (Line 105, page 6).

(2) Line 112: should the word ‘status’ actually be ‘states’ (appears twice in this line)

Thanks. We have changed the word ‘status’ to ‘states’ (Line 113-115, page 6)

(3) Line 113: change conduced to ‘conducted’

We have changed the word ‘conduced’ to ‘conducted’ (Line 113, page 6)

(4) Line 198: separates should be ‘separate’ (singular)

We have changed the ward ‘separates’ to ‘separate’ (Line 198, page 10)

(5) Line 282: “each health state” not states

We have deleted ‘s’ (Line 282, page 15)

(6) Line 328: add word to after similar- “similar to a study”

We have added a word ‘to’ (Line 328, page 17)

(7) Line 360: Indigenous Australian women (no ‘s’ necessary on Australian)

We have deleted ‘s’ (Line 361, page 18)

Reviewer #4: 

The authors have addressed a significant area of research and this article will contribute significantly towards the existing literature for cervical cancer prevention program evaluation with regards to health services uptake among indigenous communities in Australia. Yet there are some points that need to be considered:

1) Minor confusion exists in the abstract (Methods) line 40-41. Difference is not obvious between health states 2 and 3. Whereas it has been clearly indicated between S2 (HPV positive with cytology normal) and S3 (Low grade cytology) in the results section.

Thank you for pointing out this issue. Health state 2 (S2) should be ‘HPV-positive with cytology normal’. We have corrected it in Abstract and Methods sections (Line 41, page 3; and Line 140, page 7)

2) The introduction section effectively gives the background of the research problem but careful proofreading would be beneficial for language errors. Similarly the discussion section has discussed the results in comparison to previous studies, yet some language errors are present.

Thanks. We have corrected language errors:

(1). We deleted the word ‘in’ (Line 105, page 6)

(2). We added the word ‘to’ in the sentence ‘The findings are similar a study….’ (Line 329, page 17)

3) In the data collection section, the study referenced for description of sample size (19) does not explain the participants. The references need to be rearranged correctly.

We have rearranged references (Line 150, page 8)

4) Statistical analytical techniques used are adequate, tables have been formed precisely and the main findings reported appropriately. However, with my limited expertise in the field of Health Economics, I am unable to advise much.

Thanks.

---

## [Decision Letter · Decision Letter 2]

30 Jun 2021

Population-based utility scores for HPV infection and cervical squamous cell carcinoma among Australian Indigenous women

PONE-D-20-26869R2

Dear Dr. Ju,

We’re pleased to inform you that your manuscript has been judged scientifically suitable for publication and will be formally accepted for publication once it meets all outstanding technical requirements.

Kind regards,

Associate Professor Dr Muhammad Aziz Rahman

Academic Editor

PLOS ONE

Reviewers' comments:

Reviewer's Responses to Questions

**Comments to the Author**

1. If the authors have adequately addressed your comments raised in a previous round of review and you feel that this manuscript is now acceptable for publication, you may indicate that here to bypass the “Comments to the Author” section, enter your conflict of interest statement in the “Confidential to Editor” section, and submit your "Accept" recommendation.

Reviewer #2: All comments have been addressed

Reviewer #3: All comments have been addressed

Reviewer #4: All comments have been addressed

2. Is the manuscript technically sound, and do the data support the conclusions?

Reviewer #2: Yes

Reviewer #3: Yes

Reviewer #4: (No Response)

3. Has the statistical analysis been performed appropriately and rigorously? 

Reviewer #2: Yes

Reviewer #3: I Don't Know

Reviewer #4: (No Response)

4. Have the authors made all data underlying the findings in their manuscript fully available?

Reviewer #2: Yes

Reviewer #3: Yes

Reviewer #4: (No Response)

5. Is the manuscript presented in an intelligible fashion and written in standard English?

Reviewer #2: Yes

Reviewer #3: Yes

Reviewer #4: (No Response)

6. Review Comments to the Author

Reviewer #2: (No Response)

Reviewer #3: minor edits:

Line 77: full stop needs to be taken out after the [4,5]

Line 89: the full stop should be a comma

Line 101: 'and' should be 'an'

Lines 272,276,280: the inverted comma before the number 12 needs to change direction

Reviewer #4: (No Response)

7. PLOS authors have the option to publish the peer review history of their article (what does this mean?). If published, this will include your full peer review and any attached files.

Reviewer #2: No

Reviewer #3: **Yes: **Vita Christie

Reviewer #4: No

---

## [Editor Report · Acceptance letter]

14 Jul 2021

PONE-D-20-26869R2 

Population-based utility scores for HPV infection and cervical squamous cell carcinoma among Australian Indigenous women 

Dear Dr. Ju:

I'm pleased to inform you that your manuscript has been deemed suitable for publication in PLOS ONE. Congratulations! Your manuscript is now with our production department. 

Kind regards, 

on behalf of

Associate Professor Dr. Muhammad Aziz Rahman 

Academic Editor

PLOS ONE